# Bundling of Ecosystem Services in Conservation Offsets: Risks and How They Can Be Avoided

**Martin Drechsler** [1,2]

1    Helmholtz Centre for Environmental Research—UFZ, Department of Ecological Modelling, Permoserstr. 15, 04318 Leipzig, Germany; martin.drechsler@ufz.de
2    Brandenburg University of Technology Cottbus-Senftenberg, Chair of Economics, in Particular Environmental Economics, 03046 Cottbus, Germany

**Abstract:** Conservation offsets are increasingly used as an instrument to conserve biodiversity and ecosystem services on private lands. Bundling ecosystem services (ES) in the market transactions saves costs but implies that only the bundle of ES is conserved while individual ES may decline. This paper presents a simple model analysis of a conservation offset scheme to identify conditions under which bundling can lead to such undesired declines. As it turns out, these are favoured by rarity of the ES as well as a positive correlation between their abundance and the cost of their conservation. A market rule is proposed that is able to avert undesired ES declines. Rather than on sums or means of ES, this market rule focuses on the least abundant ES. Systematic variation of model parameters shows that this trading rule is most effective in those cases where the likelihood of undesired ES losses is highest.

**Keywords:** biodiversity; bundling; conservation offsets; ecosystem services; ecological–economic model





## 1. Introduction

Conservation offsets are increasingly used in many parts of the world and by now significantly influence land use on a global scale [1]. Similar to emission trading schemes, they promise to be more cost-effective than regulatory policies by allowing the trading of obligations of environmental protection [2]. While in $CO_2$ markets emission allowances are traded, in conservation offsets the object of trade is essentially biodiversity. That is, if biodiversity is destroyed at some place an equivalent amount must be created elsewhere. In practice, under a conservation offset scheme the development of ecologically valuable land requires handing in credits that are created by the restoration of degraded land. If the persons carrying out these two types of actions are not identical a market is needed on which the credits can be traded [3].

A critical issue in conservation offsets is achieving no net loss (NNL) of biodiversity [4]. The reason is that, other than $CO_2$, biodiversity is not a homogenous good (measurable by a scalar quantity) whose effect on the environment is independent of where and (within years or a few decades) when it is emitted. That means it is usually not clear whether the biodiversity lost at one site is equivalent to the biodiversity gained at the other.

In principle, equivalence can be ensured by sufficiently tight trading rules so that the biodiversity gain must be spatially very close to and in the same timeframe as the site whose biodiversity loss it is meant to offset, and the species on the restored site must be identical (and in the same abundance) to the species at the developed site. Such tight restrictions, however, would imply that the credits market could include only very few participants and the efficiency gain of the offset scheme that largely comes from number and heterogeneity of market makers would diminish [2].

Thus there is a trade-off between cost-effectiveness and equivalence [5]. The looser the trading rules the higher the cost-effectiveness of the scheme (i.e., the lower the costs of achieving NNL with respect to the biodiversity as captured by the trading rules) but the

higher the risk that—due to the missing equivalence—NNL with respect to the biodiversity as observed in the landscape is not achieved.

Trade-offs between different facets of biodiversity, or more generally between different ecosystem services (ES), are a major challenge in sustainable land management [6,7]. A prominent example in which trade-offs affect the impacts of bundling is the joint trading of biodiversity and carbon sequestration services in REDD+ (Reducing Emissions from Deforestation and Forest Degradation) schemes [8]. While there is some spatial correlation between carbon sequestration and biodiversity, so that sites rich in biodiversity also tend to sequester larger amounts of biodiversity, this correlation is not perfect, so focusing on sites that maximise total carbon sequestration does not necessarily maximise biodiversity [9]. To reduce this uncertainty one can, as an alternative to REDD+, trade biodiversity and carbon separately, but as argued above, this is likely to incur higher economic costs, not to speak of the higher transaction costs to run two separate markets rather than a single one.

Another example where bundling may lead to unwanted losses is the Sparling Ranch Conservation Bank in California analysed by [10]. Credits are awarded per hectare of land that is habitat for the California Tiger Salamander (*Ambystoma californiense*) and the California Red-Legged Frog (*Rana draytonii*). Since the scheme does not explicitly consider which of the two species is present on a land parcel, it could happen that development occurs on areas occupied mainly by one of the two species while restoration efforts benefit only the other, so that on net the former species loses habitat and declines.

Among others, the effect of bundling and in particular the risk of ES loss is likely to depend on the spatial correlations and trade-offs of the ES [11,12]. If ES are not strongly positively, or even negatively, correlated so that a large abundance in one ecosystem services implies a low abundance in others, a too strong focus on one ecosystem service risks a loss of the other—as argued for the case of the Sparling Ranch Conservation Bank.

To reduce the risk of ES loss, as a compromise between "unrestricted" bundling and the costly trading of ecosystems on separate markets, one may introduce trade restrictions within a bundling scheme. Such a restriction may be to put higher weight on the less abundant ES. In the present paper I will develop a simple simulation model of an offset scheme for the conservation of two ES to compare two options of bundling: (i) the joint abundance of both ES is considered, and (ii) only the abundance of the less abundant ES is considered. I explore the conditions under which the former design option is likely to incur net loss in one of the two ES. For the identified cases, I analyse the effectiveness of the latter design option on the observed net loss, as well as the scheme's cost-effectiveness and the trade-off between NNL and cost-effectiveness. By this the present paper contributes to a better understanding of the pros and cons of credits bundling in conservation offset schemes.

## 2. Methods

### 2.1. Why Bundling Can Incur Loss

To understand why bundling of two ecosystem services (ES) can lead to a loss in (at least) one of the two ES, consider a simple numerical example with three land parcels, indexed u, v and w (Table 1). Each land parcel can be conserved or in economic use. In the former case it leads to two ES $a$ and $b$, while in the latter case it generates an economic profit $\pi$. Change of land use is allowed as long as the sum of the two ES, $a + b$, does not decline.

Consider case I (upper left section of the table) where land parcel u is conserved, generating ES of $a = 1.5$ and $b = 1.5$, while the other two land parcels are in economic use, generating profits of $\pi = 0.5$ each. Since the (potential) economic profit $\pi = 1.5$ of the conserved land parcel u is comparatively high, its owner may wish to develop it into economic use. Considering that the sum of the two ES, $a + b$, must not decline, development of land parcel u needs to be offset by restoring the economically used land parcels v and w (right hand side of Table 1). By this, the sum of the ES remains at $a + b = 3$, while the total economic profit increases from 0.5 + 0.5 to 1.5. Thus, the transaction increases the total economic profit without reducing $a + b$. Also, $a$ and $b$ each remain at a level of 1.5, so there is loss in neither $a$ nor $b$.

**Table 1.** Example of a bundled credits trade, involving three land parcels (u, v, w; in lines) each of which can be used economically (econ) or for conservation (cons); and under economic use earns a profit $\pi$ and under conservation generates ecosystem services *a* and *b*. Numbers in parentheses give the profit or ecosystem service (ES) if the land parcel is used economically or for conservation, respectively; under the respective other land use the profit or ES, respectively, is zero. The last line shows the total profit and ecosystem services generated by the land use. Two cases, case I and case II, are considered and the left half of the table represents the situation before the land-use change and the right half represents the situation after the land-use change.

| | | Before | | | | | After | | | | |
|---|---|---|---|---|---|---|---|---|---|---|---|
| **Case I** | | $\pi$ | *a* | *b* | *a + b* | **Land Use** | $\pi$ | *a* | *b* | *a + b* | **Land Use** |
| | u | (1.5) | 1.5 | 1.5 | 3 | cons | u | 1.5 | (1.5) | (1.5) | (1.5) | econ |
| | v | 0.5 | (1) | (1) | −(2) | econ | v | (0.5) | 1 | 1 | 2 | cons |
| | w | 0.5 | (0.5) | (0.5) | −(1) | econ | w | (0.5) | 0.5 | 0.5 | 1 | cons |
| Sum | | 1 | 1.5 | 1.5 | 3 | | | 1.5 | 1.5 | 1.5 | 3 | |
| Case II | u | (1.5) | 2.5 | 0.5 | 3 | cons | u | 1.5 | (2.5) | (0.5) | (3) | econ |
| | v | 0.5 | (1) | (1) | (2) | econ | v | (0.5) | 1 | 1 | 2 | cons |
| | w | 0.5 | (0.5) | (0.5) | (1) | econ | w | (0.5) | 0.5 | 0.5 | 1 | cons |
| Sum | | 1 | 2.5 | 0.5 | 3 | | | 1.5 | 1.5 | 1.5 | 3 | |

Now consider case II. Again the owner of the conserved land parcel u wishes to develop his/her land parcel, and for this land parcels v and w need to be restored. Again, the sum of the two ES remains at *a + b* = 3 and the total economic profit increases from 0.5 + 0.5 to 1.5. However, this time *a* declines from 2.5 (generated by the conserved land parcel u before the transaction) to 1.5 (generated by the conserved land parcels v and w after the land-use change), while *b* increases from 0.5 to 1.5. The reason is that in contrast to case I, the two ES *a* and *b* are not (perfectly) positively correlated but they differ on parcel u, with *a* = 2.5 and *b* = 0.5, so that by developing that parcel more of *a* is lost than of *b*, which is not fully offset by the restoration of parcels v and w.

*2.2. The Model*

2.2.1. The Model Region

A fictitious landscape is considered with *N* = 100 land parcels with arbitrary spatial locations. Each land parcel *i* can be conserved ($x_i$ = 1) or used for economic purposes like intensive agriculture ($x_i$ = 0). Conservation of land parcel *i* delivers two ecosystem services of magnitudes $a_i$ and $b_i$. Economic use delivers no ecosystem services but generates an economic profit $\pi_i$.

Profit and ecosystem services (ES) differ among land parcels and are normally distributed. The profits have mean $m_\pi$ and standard deviation $\sigma_\pi$, and the two ES have means $m_a$ and $m_b$ and standard deviations $\sigma_a$ and $\sigma_b$, respectively (negative values are truncated to zero).

Each of the ES may be correlated with the profits. The pairwise correlations between profit and ES *a* and between profit and ES *b* are modelled using the algorithm of [13] (p. 281):

$$\pi_i = m_\pi + \sigma_\pi X_\pi$$
$$a_i = m_a + \sigma_a \left[ \rho_{\pi a} X_\pi + \sqrt{1 - \rho_{\pi a}^2} X_a \right]$$
$$b_i = m_b + \sigma_b \left[ \rho_{\pi b} X_\pi + \sqrt{1 - \rho_{\pi b}^2} X_b \right] \tag{1}$$

where $\rho_{\pi a} \in [-1, 1]$ and $\rho_{\pi b} \in [-1, 1]$ are the correlation coefficients for the correlations between $\pi$ and *a* and between $\pi$ and *b*, respectively, and $X_\pi$, $X_a$ and $X_b$ are independent normal deviates with means zero and standard deviations one.

Initially, a number of $n_0$ of (randomly chosen) land parcels are conserved and the other $N - n_0$ are in economic use. The associated initial total ES in the model landscape thus are

$$A^{(0)} = \sum_{i=1}^{N} x_i^{(0)} a_i$$
$$B^{(0)} = \sum_{i=1}^{N} x_i^{(0)} b_i. \tag{2}$$

The associated total economic profit is the sum of the profits $\pi_i$ over all economically used land parcels. Subtracting this from the maximum profit that would be obtained if all land parcels were in economic use yields the total forgone profit or total conservation cost $C^{(0)}$ of delivering the ES at levels $A^{(0)}$ and $B^{(0)}$:

$$C^{(0)} = \sum_{i=1}^{N} x_i^{(0)} \pi_i. \tag{3}$$

### 2.2.2. Dynamics of the Credits Market

The modelling of the conservation offset scheme is based on the basic concepts of tradable permit markets [14,15]. Owners of conserved land parcels with high $\pi_i$ (as land parcel u in Table 1) may wish to develop their land into economic use for which they require credits. Within the framework of a conservation offset scheme, they can buy these credits from landowners who restore economically used land parcels (for simplicity it is assumed in the present analysis that both development and restoration lead to certain and instantaneous outcomes, so there are no time lags or uncertainty and no costs associated with the land-use change: cf. [16]).

Consider for the moment a single ES $a$ and assume that the amount of credits that are required to develop a conserved land parcel is proportional to the magnitude of the lost ecosystem service. Then for owners of *conserved* land parcels $i$ with the comparatively *low* ES–cost ratios $a_i / \pi_i$ (note that $\pi_i$ can be regarded as the cost, or forgone profit, if land parcel $i$ is conserved) it will be most profitable to purchase credits to develop their land. Conversely, for landowners of *economically* used land parcels with comparatively *high* ES–cost ratios it will be most profitable to restore their land and sell the earned credits.

As a simple and stylised model of the trading process, I assume that the owner of the conserved land parcel with the lowest ES–cost ratio (land parcel u in case I of Table 1) buys credits from the owner of the economically used land parcel with the highest ES–cost ratio (land parcel w in case I of Table 1). With the exchange of the credits both landowners switch their land use.

It may happen that the gain in $a$ from the restoration of the economically used land parcels is smaller than the loss on the developed land parcel (as is the case in the example of Table 1, case I, if land parcel u is developed and only land parcel w is restored). In that case the developing landowner needs to buy additional credits, and s/he may buy theses from the landowner(s) of the economically used land parcel(s) with the next highest ES–cost ratio (land parcel v in the example of case I in Table 1)—until the loss in a from the development is fully offset by the gain from the restored land parcel.

After this first set of landowners have exchanged credits and changed their land use, there will generally be further owners of conserved land parcels who wish to develop their land. Based on the land-use pattern resulting from the actions described in the previous paragraph, I assume again that the owner of the conserved land parcel with the now lowest ES–cost ratio will exchange credits with the owner(s) of the economically used land parcels with the now highest ES–cost ratio(s). After these transactions, the owner of the conserved land parcel with the now lowest ES–cost ratio is considered, and so on, until the market has settled and there is no demand for further trading (formally, this occurs when the highest ES–cost ratio among the conserved land parcels is smaller or equal to the lowest ES–cost ratio among the economically used land parcels).

It was explained above that it may be necessary for the offsetting of a conserved land parcel's development that more than one economically used land parcels are restored, so net loss in the ES can be excluded. However, this does not exclude net gains in the ES, which would complicate a meaningful comparison of different offset schemes (cf. below). A simple way to avoid net gains is to pass any surplus that may remain from a market transaction on to the next landowner who wishes to develop his/her land. Clearly, such a surplus transfer is quite unlikely to occur in a real offset market, but such surpluses will not occur in real markets anyway and are rather an artefact of the simplicity of the present stylised model.

Now return to the case of two ES, which may be bundled as outlined in the Introduction. The simplest way to form a bundle is probably to consider the weighted sum of the ES so that in the present model, conserving land parcel $i$ would generate the bundle

$$g_i = wa_i + (1-w)b_i \tag{4}$$

where $w \in [0, 1]$ measures the importance of ES $a$ relative to that of ES $b$. Analogous to above, owners of conserved land parcels with low ES–cost ratios $g_i/\pi_i$ will demand credits to develop their land, while owners of economically used land parcels with high ES–cost ratios $g_i/\pi_i$ will restore their land and supply credits.

As outlined above, by construction of the offset scheme, the sum over the (conserved) ES, $\sum_i x_i g_i$, cannot decline in the course of the trading. However, as demonstrated in Section 2.1, no decline in the (weighted) sum of the ES does not imply no net loss in the *individual* ES, because due to Equation (4) a decline in one ES can be compensated for by an accompanying increase in the other—which in economic terms means that the two ES are perfectly substitutable.

To solve, or at least to mitigate, this problem one may restrict this substitutability between the ecosystem services during the trade. Assume the development of some land parcel $i$ leads to the loss of ecosystem services $a_i$ and $b_i$ while the restoration of some land parcels $j, k, l, \ldots$ leads to the gain of ecosystem services $a_j, a_k, a_l, \ldots$ and $b_j, b_k, b_l$, so the net gains in the two ecosystem services $a$ and $b$ are $\Delta a_{ij} = a_j - a_i$, $\Delta a_{ik} = a_k - a_i$, $\Delta a_{il} = a_l - a_i$, $\ldots$, and $\Delta b_{ij} = b_j - b_i$, $\Delta b_{ik} = b_k - b_i$, $\Delta b_{il} = b_l - b_i$, $\ldots$, each of which can be positive, zero, or negative. Under unrestricted trading (Equation (4)), the trade would be allowed if the total net gain is non-negative:

$$\Delta g_i = \left[w\Delta a_{ij} + (1-w)\Delta b_{ij}\right] + \left[w\Delta a_{ik} + (1-w)\Delta b_{ik}\right] + \left[w\Delta a_{il} + (1-w)\Delta b_{il}\right] + \ldots \geq 0 \tag{5}$$

which, as argued, does not guarantee that the total gain in *each* ES, $\Delta a_{ij} + \Delta a_{ik} + \Delta a_{il} + \ldots$ and $\Delta b_{ij} + \Delta b_{ik} + \Delta b_{il} + \ldots$, is non-negative. If, in contrast, one demands that the *minimum* of the two net gains is positive:

$$\min\{\Delta a_{ij} + \Delta a_{ik} + \Delta a_{il} + \ldots, \Delta b_{ij} + \Delta b_{ik} + \Delta b_{il} + \ldots\} \geq 0, \tag{6}$$

the non-negativity of *both* $\Delta a_{ij} + \Delta a_{ik} + \Delta a_{il} + \ldots$ and $\Delta b_{ij} + \Delta b_{ik} + \Delta b_{il} + \ldots$ is guaranteed. The trade under this rule is modelled analogously to the trade under Equation (4), so that owners of conserved land parcels with low ratios

$$\frac{g_i\prime}{\pi_i} \equiv \frac{\min\{a_i, b_i\}}{\pi_i} \tag{7}$$

will demand credits to develop their land, while owners of economically used land parcels with high ratios $g_i'/\pi_i$ will restore their land and supply credits.

To assess the impact of each of the two trading rules, the dynamics of trading and land-use change are modelled as described above, which result after the market has settled in a new land-use pattern $\mathbf{x}^{(1)} \equiv (x_1^{(1)}, x_2^{(1)}, \ldots, x_N^{(1)})$.

The associated total ES and total conservation cost (forgone economic profit) are

$$
\begin{aligned}
A^{(1)} &= \sum_{i=1}^{N} x_i^{(1)} a_i \\
B^{(1)} &= \sum_{i=1}^{N} x_i^{(1)} b_i \\
C^{(1)} &= \sum_{i=1}^{N} x_i^{(1)} \pi_i.
\end{aligned}
\tag{8}
$$

### 2.3. Model Analysis

The aim is to compare the performances of the two offset schemes implied by the credit trading rules of Equations (4) and (6). In a first step I calculated for the trading rule of Equation (4) the relative gains (or losses) in the ES $A$ and $B$ and the conservation cost $C$:

$$
\Delta A \equiv 1 - \frac{A^{(1)}}{A^{(0)}}, \ \Delta B \equiv 1 - \frac{B^{(1)}}{B^{(0)}}, \ \Delta C \equiv 1 - \frac{C^{(1)}}{C^{(0)}}.
\tag{9}
$$

I am, in particular, interested in the conditions under which $\Delta A$ is likely to be negative, so there is a loss in $A$ (since the model is "symmetric" in the two ES, the results for $B$ are identical). For this I built 10,000 random model parameter combinations in which each model parameter is drawn from its range given in Table 2.

**Table 2.** Ranges of the model parameters.

| Model Parameter | Symbol | Range |
|---|:---:|:---:|
| Initial proportion of conserved land parcels | $n_0$ | [0, 0.5] |
| Mean profit | $m_\pi$ | 1 |
| Mean ecosystem service $A$ | $m_a$ | [0.25, 4] |
| Mean ecosystem service $B$ | $m_b$ | [0.25, 4] |
| Coefficient of variation profit | $CV_\pi$ | [0, 0.3] |
| Coefficient of variation ecosystem service $A$ | $CV_a$ | [0, 0.3] |
| Standard deviation ecosystem service $B$ | $CV_b$ | [0, 0.3] |
| Correlation between profit and ES $A$ | $\rho_{\pi a}$ | [−1, 1] |
| Correlation between profit and ES $B$ | $\rho_{\pi b}$ | [−1, 1] |
| Weight of ES | $w$ | [0, 1] |

By choosing $m_\pi = 1$, all economic quantities (profits and (total) conservation costs) are effectively scaled in units of $m_\pi$. The means $m_a$ and $m_b$ span a range between considerably smaller than and considerably larger than one (values were drawn from this range on a log scale, so the likelihood of sampling a value, e.g., between 0.5 and 1 is equal to the likelihood of sampling a value between 1 and 2 or between 2 and 4). A coefficient of variation of $CV = 0.3$ implies that the ratio of the upper and lower 5% quantiles of drawn numbers is about $(1 + 2 \times 0.3))/(1 - 2 \times 0.3) = 4$ which represents quite a large variation. The ranges for $\rho_{\pi a}$, $\rho_{\pi b}$ and $w$ cover what is mathematically possible, while $n_0 > 0.5$ is rather unlikely to occur in real conservation problems.

The following analyses were carried out:

1.  For each of the 10,000 model parameter combinations I calculated $\Delta A$ to obtain a vector with elements $\Delta A_s$ ($s = 1, \ldots, 10{,}000$). This vector is correlated with each of the nine vectors composed of the 10,000 values of the nine model parameters (excluding $m_\pi = 1$). A large (Pearson) correlation coefficient near +1 indicates (cf. [17]) that an increase in the focal model parameter increases $\Delta a$, i.e., reduces the likelihood of a loss in ES $a$, while a small value near −1 indicates that the model parameter increases the likelihood of a loss.

2.  Then I split the range of each model parameter in thirds and determined the third that is associated with the highest likelihoods of net loss. Considering this third for each of the model parameters, the parameter space is restricted to those values of the model

parameters where a loss in *a* is most likely. Similar to above I drew 10,000 random model parameter combinations from this restricted parameter space and determined the mean of the $\Delta A_s$ for the trading rule of Equation (4) and the trading rule of Equation (6). For each trading rule I further calculated for each model parameter combination *s* the benefit–cost ratios $A^{(1)}{}_s/C^{(1)}{}_s$ and $B^{(1)}{}_s/C^{(1)}{}_s$ and took the mean over all *s*. These means measure, for each of the two trading rules, the scheme's mean cost-effectiveness with respect to the conservation of the two ES *a* and *b*.

3. Lastly, for each model parameter combination *s* (in the restricted parameter space) I subtracted $\Delta A_s$ with trading rule Equation (6) from $\Delta A_s$ with trading rule Equation (4) to determine the impact of the trading rules on the loss in ES *a*. Analogous to step 1 I correlated these 10,000 differences to the values of the nine model parameters to determine under which conditions a change of the trading rule from Equations (4)–(6) has the strongest effect on the loss in *a*.

## 3. Results

### 3.1. Impact of Model Parameters on the Net Loss in ES a

Table 3 shows the correlations between the loss $\Delta A$ and the nine model parameters. A positive correlation indicates that an increase (decrease) in the corresponding model parameter increases (decreases) $\Delta A$; while a negative correlation indicates that an increase (decrease) in the corresponding model parameter decreases (increases) $\Delta A$. By this, $\Delta A$ tends to be large if $n_0$ is large, $m_a$ is small, $m_b$ is large, $CV_\pi$ is small, $CV_a$ is small, $CV_b$ is large, $\rho_{\pi a}$ is large, $\rho_{\pi b}$ is small, and $w$ is small.

**Table 3.** Pearson correlation coefficients between model parameter and net loss $\Delta A$.

| Model Parameter | Symbol | Correlation |
|---|---|---|
| Initial proportion of conserved land parcels | $n_0$ | 0.42 |
| Mean ecosystem service A | $m_a$ | −0.07 |
| Mean ecosystem service B | $m_b$ | 0.08 |
| Coefficient of variation profit | $CV_\pi$ | −0.17 |
| Coefficient of variation ecosystem service A | $CV_a$ | −0.17 |
| Coefficient of variation ecosystem service B | $CV_b$ | 0.10 |
| Correlation between profit and ES A | $\rho_{\pi a}$ | 0.02 |
| Correlation between profit and ES B | $\rho_{\pi b}$ | −0.20 |
| Weight of ES | $w$ | −0.16 |

### 3.2. Loss in ES a and Scheme Cost-Effectiveness under the Most Adverse Conditions

Based on the conclusions from Table 3, I built 10,000 random parameter combinations that favoured large $\Delta A$ by sampling $n_0$ from the upper third of the original interval (Table 2), i.e., from [0.333, 0.5], $m_a$ from [$2^{-2}$, $2^{0.333}$], $m_b$ from [$2^{0.333}$, $2^2$], $CV_\pi$ from [0, 0.1], and so on. For these parameter combinations, the trading rule of Equation (4) leads to an average loss of 0.06, which changes into a gain of 0.07 if the trading rule is replaced by Equation (6) (Table 4). Similarly, changing the trading rule from Equations (4)–(6) increases the cost-effectiveness, measured by the mean of $A/C$, with respect to ES *a* from 039 to 0.42. Conversely the cost-effectiveness with respect to ES *b* declines from 2.83 to 2.62. With respect to the conservation of the bundle A + B, the cost-effectiveness declines from 3.21 to 3.04.

**Table 4.** Mean of ecosystem services loss $\Delta A$ over 10,000 model parameter combinations, and mean of the benefit–cost ratios $A/C$ and $B/C$ $(A + B)/C$ for the ES $a$ and $b$. Obtained for the two credit trading rules of Equation (4) and Equation (6), respectively.

| Trading Rule | Quantity | Value |
|:---:|:---:|:---:|
| Equation (4) | Mean of $\Delta A$ | 0.06 |
| | Mean of $A/C$ | 0.39 |
| | Mean of $B/C$ | 2.83 |
| | Mean of $(A + B)/C$ | 3.21 |
| Equation (6) | Mean of $\Delta A$ | −0.07 |
| | Mean of $A/C$ | 0.42 |
| | Mean of $B/C$ | 2.62 |
| | Mean of $(A + B)/C$ | 3.04 |

*3.3. Impact of Model Parameters on the Effect of the Trading Rules*

The effect of the trading rule was measured (for each of the 10,000 model parameter combinations in the restricted parameter space used in Section 3.2) by subtracting $\Delta A$ obtained with the trading rule of Equation (6) from $\Delta A$ obtained with the trading rule of Equation (4). Since $\Delta A$ of Equation (6) is (on average) smaller than $\Delta A$ of Equation (4) I multiplied these differences with −1 (effectively subtracting $\Delta A$ of Equation (4) from $\Delta A$ of Equation (6)) to obtain (on average) positive numbers that measure the averted loss achieved when changing from Equations (4)–(6).

The correlation of this difference (over the 10,000 model parameter combinations) with the nine model parameters is shown in Table 5. A positive correlation means that a large (small) value of the corresponding model parameter favours a high (low) amount of averted loss; while a negative correlation means that a large (small) value of the corresponding model parameter favours a low (high) amount of averted loss. Table 5 then indicates that aversion of net loss by changing from Equations (4)–(6) is highest if $n_0$ is small, $m_a$ is small, $m_b$ is large, $m_\pi$ is large, $CV_a$ is small, $CV_b$ is large, $\rho_{\pi a}$ is large, $\rho_{\pi b}$ is small, and $w$ is small.

**Table 5.** Pearson correlation coefficients between model parameter and the loss $\Delta A$ averted when the trading rule is changed from Equations (4)–(6).

| Model Parameter | Symbol | Correlation |
|:---:|:---:|:---:|
| Initial proportion of conserved land parcels | $n_0$ | −0.30 |
| Mean ecosystem service $A$ | $m_a$ | −0.14 |
| Mean ecosystem service $B$ | $m_b$ | 0.12 |
| Coefficient of variation profit | $CV_\pi$ | 0.02 |
| Coefficient of variation ecosystem service $A$ | $CV_a$ | −0.08 |
| Coefficient of variation ecosystem service $B$ | $CV_b$ | 0.51 |
| Correlation between profit and ES $A$ | $\rho_{\pi a}$ | 0.30 |
| Correlation between profit and ES $B$ | $\rho_{\pi b}$ | −0.14 |
| Weight of ES | $w$ | −0.36 |

## 4. Discussion

The bundling of ecosystem services (ES) in conservation offset schemes has considerable advantages, since it reduces transaction costs (cf. [3,18,19]) and reduces the risk of market participants by allowing compensation for the loss in one ES by a corresponding gain in others [20]. However, the latter economic advantage can be an ecological disadvantage, because no net loss in the bundle does not guarantee no net loss in each individual ES. The reason is that the bundle is usually defined as the sum (or mean) of the individual ES, so that the reduction of one ES can be compensated for by an increase in the another.

In this paper a simple generic model of a conservation offset scheme was developed in which two ES are bundled, so that market participants effectively trade a weighted sum (or mean) of the individual ES. The model analysis showed that the loss of individual ES (despite no loss in the mean) is favoured especially if the focal ES is less abundant than the

other, if it is assigned a smaller weight by the conservation manager than the other ES, and if it is positively correlated with the costs of conserving this ES, while the other ecosystem service is negatively correlated with the costs (Table 3). The latter result is because the landowners base their decisions on the ES–cost ratios. This implies that eventually the most costly (or economically profitable) land parcels tend to be used economically while the least profitable ones tend to be conserved, so the positively correlated ES declines while the negatively correlated one increases.

To avoid such unwanted losses of individual ES, one may base the credits associated with a bundle on the less abundant ES. Indeed, especially in the described cases that favour declines of individual ES, this "minimum trading rule" quite effectively reduces the risk of ES loss. A disadvantage of this trading rule is that it may raise the overall costs of ecosystem conservation (Table 4).

The minimum rule is a limiting case of the complementarity concept used in economics. Complementarity means that the value of one good increases with the abundance of the other good(s). Or conversely, if one good becomes very rare the values of the other goods decline, as well. By this, with increasing the level of complementarity the value of the bundle of goods is increasingly determined by the abundance and value of the least abundant good—with the extreme that the value of the bundle is entirely determined only by the least abundant good. An avenue for future research may be to generalise the present minimum trading rule to a rule based on complementarity. In addition, the analysis may be expanded to consider the spatiotemporal dynamics of ES as well as other complexities of ecosystems such as feedback loops, time lags, and nested phenomena [20].

Nevertheless, the present analysis provides some first helpful insights into the consequences of ecosystem bundling in conservation offset schemes and how unwanted losses of individual ES may be avoided. The analysis is based on a simple and stylised model, which however captures some essential elements of a real landscape. It may be regarded as a starting point for more complex theoretical or empirical analysis of the implications of bundling in conservation offset schemes and the prevention of undesired consequences.

**Funding:** This research received no external funding.

**Institutional Review Board Statement:** Not applicable.

**Informed Consent Statement:** Not applicable.

**Data Availability Statement:** The data presented in this study are available on request from the author.

**Conflicts of Interest:** The author declares no conflict of interest.

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
