# Peer review of "Bundling of Ecosystem Services in Conservation Offsets: Risks and How They Can Be Avoided"

_land, doi:10.3390/land10060628_

Round 1

Reviewer 1 Report

Dear authors, This manuscript constitutes an interesting and well performed study about bundling of ecosystem services in conservation offsets, the risks and how they can be avoided. The overall idea is interesting, and the methodology is very well and extensively described. The manuscript, in general, is clear and well-structured. Discussion is very short and limited, with not any references and comparisons with examples relative to the subject and the results of the paper and needs to be updated and enriched with references concerning ecosystem services and relative studies.

Author Response

Response: Thank you for the favourable comments. I agree there were not enough references to other work in the Discussion. I rewrote the first paragraph of the Discussion section and expanded the second last paragraph to better relate the present work to previous works.

Reviewer 2 Report

Dear all,

Thank you for the opportunity to read this paper.

The aim of this study is scientifically interesting and useful. It presents a model analysis to identify conditions under which bundling ecosystem services can lead to such undesired declines and proposes a market rule that is able to avert them.

The work is well structured nevertheless, I think some small effort should be made to improve it in the discussion of the results.

Specifically, the below aspects should be addressed by the author before publication.

SPECIFIC COMMENTS:

Abstract: I think an extra effort should be made to briefly explain the methodology and tools used in the study and what kind of results were achieved.

Methodology: Is this methodology a novelty or has it already been used in other work?

Section 2.2.2. “As a simple and stylised model of the trading process, I assume..” Please, could you avoid personal expressions such as “I assume” or “I am interested” etc.

Table 3: Could you better explain and discuss this table results?

Line 320 – 326: I think this sentence is a bit too obvious and weak.

Author Response

The aim of this study is scientifically interesting and useful. It presents a model analysis to identify conditions under which bundling ecosystem services can lead to such undesired declines and proposes a market rule that is able to avert them.

The work is well structured nevertheless, I think some small effort should be made to improve it in the discussion of the results.

Response: Thank you for the positive comments. The final comment goes into the same direction as Reviewer 1’s comment: I rewrote the first paragraph of the Discussion section and expanded the second last paragraph to better relate the present work to previous works.

Specifically, the below aspects should be addressed by the author before publication.

Specific comments:

Abstract: I think an extra effort should be made to briefly explain the methodology and tools used in the study and what kind of results were achieved.

Response: I expanded the Abstract to include more information about methodology and specific results.

Methodology: Is this methodology a novelty or has it already been used in other work?

Response: It is based on basic concepts of tradable permits. I included this information together with two references.

Section 2.2.2. “As a simple and stylised model of the trading process, I assume..” Please, could you avoid personal expressions such as “I assume” or “I am interested” etc.

Response: Although those “I’s” are personal statements, to my knowledge active voice should be preferred to passive voice; and in these particular cases I also feel the present formulation makes clear that they refer to the present author and not to some previous research. So I decided to leave these expressions as they are.

Table 3: Could you better explain and discuss this table results?

Response: Thank you for this comment. The description of Table 3 is in the first paragraph of section 3.2. I suppose some confusion arose because I wrongly labelled Table 4 as Table 3, too; and also wrongly referred to it as Table 3 in the text (second paragraph of section 3.3). I fixed these bugs and hope that both Tables 3 and 4 are understandable now.

Line 320 – 326: I think this sentence is a bit too obvious and weak.

Response: I agree and rewrote this sentence (now included in the rewritten first paragraph of the Discussion section).

Round 2

Reviewer 1 Report

Dear author,

I think your manuscript can now be accepted for publication.

Please check References(after reference No19, there are some more not numbered).